# Which Protective Factors Are Associated with the Mental Health of Syrian Students in Germany? A Register-Based Cross-Sectional Study

**DOI:** 10.3390/ijerph192316200

**Published:** 2022-12-03

**Authors:** Remy Rahim Hosari, Andrea Borho, Eva Morawa, Yesim Erim

**Affiliations:** Department of Psychosomatic Medicine and Psychotherapy, Friedrich-Alexander-University Erlangen-Nürnberg (FAU), 91054 Erlangen, Germany

**Keywords:** depression, generalized anxiety disorder, mental health, post-traumatic stress disorder, protective factors, Syrian students, university

## Abstract

The aim of this online, register-based cross-sectional study was to investigate the frequency of psychological distress and protective factors among 136 Syrian students in Germany. The survey measured depression and anxiety (Patient Health Questionnaire-4), post-traumatic stress (Impact of Event-Scale-6), as well as resources social support (ENRICHD Social Support Instrument), optimism (Optimism–Pessimism-2 Scale) and religiosity (Duke Religion Index). A total of 26.50% of the sample were female. A total of 38.93% met criteria for clinically relevant depressive respectively generalized anxiety symptoms and 15.72% showed prominent PTSD scores. Participants screened positive for mental distress reported significantly less social support (*p* = 0.001) and less optimism (*p* = 0.002) than participants without mental distress. In multiple regression analyses, higher levels of feeling welcome in Germany, social support and intrinsic religiosity were significantly associated with lower levels of depression and generalized anxiety. Significant associations with lower PTSD levels were found with higher levels of social support and optimism. The results show that Syrian students in Germany are more psychologically burdened compared to other Syrian refugee samples, except for PTSD. This suggests that besides the stress caused by flight and trauma, stressors such as studying and social isolation could be considered as additional impediments for mental health and require intervention measures.

## 1. Introduction

Syria prided itself on having one of the most modern and largest higher education systems in the Middle East. From 2001 to 2010, a number of reforms were carried out to improve the higher education system, such as equipping it with modern laboratories and introducing biotechnology. In addition, close cooperation was established with international research groups [1].

With the emergence of the civil war in Syria, massive impediments and shortages emerged in life in general and in the school and higher education system in particular. Since 2011, up to 300,000 Syrians have been killed and half the population was displaced. The personal safety of the citizens was very critical, which was also reflected in the safety of university staff and students at the University campus [2]. All this has left great damage on the scientific level, such as stagnation, limited internationalization and the disappearance of research [3], as well as the mass exodus of the teaching staff who left the country [2].

All these difficult conditions led Syrian students to leave their homeland for better lives in other corners of the world. However, access to higher education in the host countries for the Syrian students was more difficult than expected. Due to missing documents, such as birth certificates and school reports, the process of their integration at other universities around the world was made more difficult [4].

In addition to missing documents, the legitimation of Syrian certificates was not an easy issue in many of the host countries. On the one hand, some important documents were lost in the turmoil of war, and on the other hand, previous school and university achievements could not always be recognized. Consequently, the number of Syrian students who could be registered at a university was relatively small. In Germany, that number was 19,366 in 2021, while 818,000 Syrians lived in Germany in 2020 [5].

Several studies in their host countries show that refugees exhibit high prevalences of posttraumatic stress disorder (PTSD), anxiety and depression, in some instances even several times more often than the general population [6,7,8,9]. A prospective study from our working group examined prevalence of those entities in Syrian refugees over two measurement points, over an average of 1.5 years after arrival in Germany. It found that 27% of the participants suffered from depression at the first measurement, 18% from generalized anxiety disorder and about 30% of these 108 participants suffered from PTSD [6].

The loss and separation from loved ones, as well as longing for their home country, were identified as the cause of psychological distress [10]. Factors affecting the mental health, life satisfaction and quality of life of Syrian refugees were examined in detail by Al Masri et al. [11].

Postmigration stressors have received closer attention in recent years. Numerous studies have shown that postmigratory stressors among refugees actually elucidate greater variance in levels of depression and anxiety than war-related trauma and loss experiences [12,13,14,15]. They have also been positively associated with post-traumatic stress disorder [13,15,16]. In their widely cited work, Laban et al. [17] summarized postmigration stressors into five categories: family-related aspects, discrimination, asylum procedures, socio-economic living conditions and socio-religious aspects. These clusters were significantly related to depressive disorders.

Additionally, several protective factors, such as mental stability or resilience, have been explored in refugee populations. Protective factors in clinical psychology is a term that summarizes factors that reduce the probability of the occurrence of mental disorder in the presence of psychological distress. Protective factors can be distinguished in individual and environmental attributes that are associated with positive adjustment and development throughout the course of life-threatening conditions and cultural situations [18].

The role of resilience or sense of coherence only comes into play after a longer stay in the host country, but in the first stages of adaptation, specific skills such as language competence have a greater impact on coping [19]. Trait resilience moderated the effects of trauma exposure on PTSD severity, with higher resilience levels attenuating the effect of traumatic exposure on PTSD development (Fino et al. 2020) [20]. In Syrian youths (ages 13–24), resilience was significantly associated with health-related quality of life and general mental distress but not with PTSD [21].

In addition, religiosity, as a protective factor, has been studied in more detail in several studies. US researchers demonstrated in a qualitative study among 10 refugee participants that religion played an important role in giving people hope for the future. Many participants shared how they used faith to support themselves in difficult times and to gain hope that their situation would improve [22,23]. The topic of religiosity and its influence on health has also interested German researchers. In a cross-sectional study involving 257 first-generation immigrants of Polish origin living in Germany, Morawa et al. showed that intrinsic religiosity acts as a protective factor against some cardiovascular risk factors (smoking and alcohol consumption) among Polish immigrants [24]. In addition to religion, optimism and social support are important protective factors among refugees. Studies have shown that optimism correlates strongly with post-traumatic growth (PTG; positive change in emotional and cognitive life counteracting PTSD), whereby optimism was not a statistically significant predictor of PTSD, but was a predictor of PTG [25,26]. Other studies have focused on the importance of social support. For example, a Swedish study with Syrian refugees showed that those who had higher levels of social support were less likely to suffer from PTSD than those who reported lower levels of social support [27]. A study by Dehnel and colleagues [28] investigated the mental health of Syrian refugees, but the focus was on children, more specifically, the influence of resilience on mental health.

In sum, the prevalence of mental disorders [6,7,8,9] and protective factors among Syrian refugees have been well studied [28,29,30,31], but there is a lack of detailed studies on the situation of Syrian students in particular. Students represent a high-risk population because, in addition to the stressors of social integration, they must adapt to studying in a highly efficient and performance-based system, mostly in a newly acquired language.

Therefore, the aim of our study was to investigate the frequency of mental disorders among Syrian students and the related protective factors. Another aim was to compare subjects with vs. without clinically relevant mental stress symptoms regarding their protective factors. On the basis of these objectives, we made two hypotheses regarding the protective factors and mental health of our subjects.
Participants with clinically relevant symptoms of mental distress experience less social support, are less optimistic and are less religious than participants without mental distress.Religiosity, social support and optimism are significantly negatively associated with mental disorders such as depression, anxiety disorder and PTSD.


## 2. Materials and Methods

This investigation was designed as a cross-sectional register-based online study on resources and protective factors of mental health among Syrian students in Germany. It was performed as an online survey for the Syrian students of the Friedrich-Alexander University Erlangen-Nürnberg (FAU). The survey was programmed in German and Arabic language with the academic online survey tool Unipark (www.unipark.com (accessed on 26 February 2021). We contacted students through their university emails, which we had received from the university administration and sent out 238 participation invitations with high priority in March 2021. Participants who did not respond to the first invitation were sent four reminders at weekly intervals. 136 students (57%) were successfully recruited, of which 129 (92%) students completed the questionnaire to the end. Study inclusion criteria were aged above 18, Syrian nationality and being a student at the Friedrich-Alexander University Erlangen-Nürnberg (FAU) in the winter semester 2020/2021. After being informed about the study course and goals, participants gave their online consent and completed the questionnaires. Participation was anonymous and voluntary, and all of the study participants received a reimbursement of a shopping voucher over 15 €. In order to receive the shopping voucher, participants were asked at the end of the survey to go to a link where they could leave their email address. The voucher was then sent to this email address. The email address was saved independently of the information in the questionnaire and could not be assigned at any time. The study was approved by the Ethics Committee of the Medical Faculty of the Friedrich-Alexander University Erlangen-Nürnberg (FAU) (project identification code: 74_17B).

### 2.1. Assessment Instruments

This survey included sociodemographic and migration-related variables, as well as depressive symptoms, symptoms of generalized anxiety disorder and symptoms of post-traumatic stress disorder. In addition, protective factors such as social support, optimism, religiosity and resilience were assessed. Other protective factors such as life satisfaction and being welcome in Germany were measured with the help of rating scales. The questions of the survey were translated by two independent translators from German into Arabic, back translated and finally combined into a final version. The comprehensibility and cultural validity of the assessment instruments were previously tested.

#### 2.1.1. Depression and Generalized Anxiety Disorder: PHQ-4 (Patient Health Questionnaire)

PHQ-4 [32] as an abbreviated form of the Patient Health Questionnaire (PHQ-D) was used to investigate depression and generalized anxiety symptoms among the Syrian students. This ultrashort form of the PHQ-D can be utilized as a total score or subdivided into a module for depression (PHQ-2) and a module for generalized anxiety symptoms (GAD-2). For this sample, the Cronbach’s Alpha was 0.8 for PHQ-2 and 0.8 for GAD-2.

##### Depression (PHQ-2)

The PHQ-2 [33] inquired about the frequency of depressed mood and anhedonia over the past two weeks. The purpose of the PHQ-2 is to screen for clinically relevant depressive symptoms in a “first-step” approach. Participants reported, on a Likert scale from “not at all” (0) to “almost every day” (3), how often they felt “down, depressed or hopeless” and had “little interest or pleasure in doing things” in the last two weeks. PHQ-2 score was obtained by adding the scores for each question (total points). A cut-off value of ≥3 was described as optimal for screening.

##### Generalized Anxiety Disorder (GAD-2)

The Generalized Anxiety Disorder Scale 2 (GAD-2) [34] is an ultrashort and easy to complete initial screening tool for clinically relevant generalized anxiety disorder symptoms. Severity is rated on a 4-point Likert scale ranging from “not at all” (0) to “present nearly every day” (3) during the past two weeks. Sum scores could be assigned to the severity of GAD symptoms. GAD-2 score was obtained by adding score for each question (total points). A score of 3 points was the cut-off for identifying possible cases in which further diagnostic evaluation for generalized anxiety disorder is recommended. Using a cut-off of 3, GAD-2 has a sensitivity of 86% and specificity of 83% for the diagnosis of generalized anxiety disorder [34].

#### 2.1.2. Post-Traumatic Stress Disorder (IES-6)

The Impact of Event Scale-6 (IES-6) is a self-assessment questionnaire for measuring post-traumatic stress disorder reactions, recording the occurrence of psychological reactions in the last seven days after a traumatic event. It is an abbreviated six-item-version of the IES-R [35]. The six items are added up to a total score. Respondents are asked to report how distressed or bothered they are, by symptoms related to a specific trauma, using the following scale: “not at all” (item score 0), “a little bit” (score 1), “moderately” (score 2), “quite a bit” (score 3), or “extremely” (score 4). We calculated a Cronbach’s Alpha value of 0.8. As there is no validated version of the German IES-6, we calculated the cut point via Youden Index [36].

#### 2.1.3. ENRICHD Social Support Instrument

The ENRICHD Social Support Instrument (ESSI) was used to assess the level of emotional social support in the participants’ lives [37]. The perceived social support was measured using a five-point response scale (0, never; 1, rarely; 2, sometimes; 3, most of the time; and 4, always). The individual items were then added up to a total score, whereby higher scores signifying stronger social support. The reliability of the ESSI using Cronbach’s Alpha equaled 0.9.

#### 2.1.4. Intrinsic Religiosity (DUREL)

The Duke Religion Index [38] is a five-item scale that captures three major dimensions of religiousness: organizational, non-organizational and subjective or intrinsic religiosity. The first two items were taken from large community and clinical studies conducted in North Carolina. The final three items were extracted from Hoge’s 10-item intrinsic religiosity scale [39]. The resulting index captures three dimensions of religiousness that are related in overlapping yet unique ways to social support and different health outcomes [38]. Given that it is not recommended to add all subscales to a total score, we only calculated the sum score for intrinsic religiosity. Higher values mean higher religiosity. Cronbach’s Alpha amounted to 0.9.

#### 2.1.5. Optimism–Pessimism-2 Scale (SOP2)

The SOP2 (9) is an economic scale to measure the psychological characteristic optimism–pessimism. The scale is easy to administer in different survey modes. As we are interested in the protection factor optimism, we asked the first item: “Optimists are people who look to the future with confidence and expect good things most of the time”: Please rate yourself: How optimistic are you in general?” A seven-point rating scale was available for the respondents with answers ranging from “not at all optimistic” (1) to “very optimistic” (7).

### 2.2. Statistical Analysis

Statistical analysis was conducted using IBM SPSS Statistics 28 (IBM Corporation, Armonk, NY, USA). To profile the socio-demographic and migration characteristics of the sample, the following descriptive statistics were calculated: means, standard deviations, ranges and frequencies. The prevalence rates of the assessed mental disorders were calculated based on the available cut-off values for each questionnaire. For the analysis of the first hypothesis, we compared participants screened positive for mental distress with those screened negative for mental distress regarding their reported optimism, social support and religiosity using *t*-tests for independent samples for continuous variables. For the analysis of the second hypothesis, wherein religiosity, social support and optimism are significantly associated with depressive, anxiety and PTSD symptoms, multiple linear regressions were calculated. There were no correlation coefficients greater than 0.70, so none of the predictor variables tested had to be excluded from multiple regression models due to multicollinearity. The significance level was set at *p* ≤ 0.05 in all analyses.

## 3. Results

### 3.1. Sample Characteristics (Socio-Demographic and Migration-Specific Variables)

The sample included 136 participants who are registered at the Friedrich-Alexander University Erlangen-Nürnberg and have Syrian nationality.

Thirty-six of the participants were female (26.50%). The mean age of the participants was 25.89 years (SD = 3.42, range: 19–37). Among all participants, six (4.42%) were parents.

The majority of participants (60.54%) had their residence permit valid for 1–2 years. Further sociodemographic and migration-specific variables are shown in Table 1.

### 3.2. Life Satisfaction in Germany

On a visual scale from “not satisfied at all” (0) to “very satisfied” (10), satisfaction with life in Germany was reported to be relatively high, averaging M = 6.85 (SD = 2.48; range: 0–10).

### 3.3. The Prevalence of Depression, Generalized Anxiety Symptoms and PTSD

The total score for depression was M = 2.34 (SD = 1.9; range: 0–6). A total of 51 participants (38.93%) were found to have clinically relevant depressive symptoms.

For generalized anxiety disorder, there were also 51 participants (38.93%) with clinically relevant anxiety. The calculated average for the total sample amounted to M = 2.41 (SD = 1.99; range: 0–6).

Relevant symptoms of post-traumatic stress disorder were reported by 18 participants (15.72%) with a score of M = 1.13 (SD = 0.65; range: 0–3) for the total sample.

### 3.4. Optimism, Social Support and Religiosity: Comparison between Subjects with vs. without Clinically Relevant Symptoms

Optimism among the study participants had a mean value of M = 5.26 (SD = 1.39; range: 1–7). A significant difference (*p* = 0.002 concerning optimism levels was observed between the group of mentally disordered and non-mentally disordered subjects. Regarding social support, our subjects had a mean sum score of M = 18.54 (SD = 5.14; range: 5–25), and the comparison revealed higher values among non-mentally disordered students (M = 19.79; SD = 4.52) than mentally disordered participants (M = 16.95; SD = 5.43; *p* = 0.001). Intrinsic religiosity showed a mean sum value of M = 8.23 (SD = 3.26; range: 3–12). The comparison between the two groups showed no significant difference. More detailed information can be found in Table 2.

### 3.5. Predictors of Depression, Generalized Anxiety and PTSD Symptoms

Multiple regression analyses were performed to investigate the potential correlation between various socio-demographic and migration-specific variables as well as other protective factors, such as social support and the severity of depression, generalized anxiety, and PTSD symptoms (Table 3).

Female gender (β = −0.18, *p* = 0.02), and higher age (β = 0.26, *p* < 0.001) were significantly related to increased symptoms of depression. In addition, a stronger feeling of being welcome in Germany (β = −0.28, *p* < 0.001), higher social support (β = −0.18, *p* = 0.03), and higher religiosity were associated (β = −0.23, *p* = 0.002) with lower depressive symptoms. The explained variance was 37.80% (F (7, 121) = 10.49, *p* < 0.001).

For generalized anxiety disorder symptoms, a significant negative relation with religiosity (β = −0.18, *p* = 0.02), with the feeling of being welcome in Germany (β = −0.24, *p* = 0.004), as well as with social support (β = −0.24, *p* = 0.007), were detected. The explained variance was 30.8% (F (7,121) = 7.70, *p* < 0.001).

Regarding PTSD, the significant predictors were social support (β = −0.35, *p* < 0.001) and optimism (β = −0.19, *p* = 0.05) with an explained variance of 23.5% (F (7,105) = 4.60, *p* < 0.001).

## 4. Discussion

As far as we know, this is the first study to investigate the mental health of Syrian students registered at a German university.

Regarding the sociodemographic data analysis, out of 136 participants, 73.5% male subjects participated in our study. This roughly corresponds to the data of other studies [6,11]. One possible explanation is that a large proportion of young Syrians who leave Syria and come to Germany tend to be young men. For the same reason, the age structure of our sample and other samples is very similar [6,11].

Our findings revealed that Syrian students were relatively well satisfied with their life in Germany, and the results are comparable to the results of studies on the general Syrian refugee population [6,11]. However, the prevalence of mental disorders was slightly higher in our sample. For example, in the study by Borho and colleagues [6], 18% suffered from generalized anxiety disorder and about 30% of these 108 participants suffered from PTSD, while in our sample the values were 38.93% for depression and generalized anxiety disorder, respectively, and 15.72% for PTSD. One possible explanation for the manifestation of higher depression and anxiety among students could be the requirements to be fulfilled in order to be admitted to a degree programme or the stress during college studies because non-native speakers usually have to put in more effort, especially at the beginning of their studies, to achieve similar results to their German-speaking fellow students [40]. However, a closer look at the values shows that the frequency of PTSD is lower in our sample. This can be explained by the fact that 14% of the students came to Germany by student visas and have not experienced a flight journey, as was the case with many Syrians in other study samples. Therefore, they were not exposed to the violent and hazardous events a flight journey usually entails.

In our sample, age was observed to be positively associated with depression. A possible explanation for this tendency among Syrians in Germany is the fact that the ability to adapt decreases with age. Learning a new language and a new rhythm of life may make the integration process more difficult, thereby leading to a feeling of loneliness or longing for the old life back home. Concerning gender, our results were consistent with the results of other studies. Borho et al. [6] found that female gender correlated significantly with more severe depression symptoms, as well as generalized anxiety disorder and PTSD. This could be due to the fact that female participants are more intensely concerned with their mental health, whereas male participants tend to assess their health as generally good. Finally, also a very important variable that correlates with the mental health parameters of Syrian students is the feeling of being welcome in Germany. This variable was not found in other studies but has been added because it is an often-named issue in the public discussion, and finally, it had a significant effect in our sample. There were significant negative correlations with depression and with generalized anxiety symptoms. It can be assumed, that people who feel welcome feel socially accepted and socially included. This leads to students feeling less lonely, less alien and less frightened in Germany. As a result, the associated consequences, such as depression and generalized anxiety disorder, are less likely to occur [41]. This result also matches with previous studies having shown that discrimination is negatively associated with mental health (e.g., Viazminsky et al. 2022) [41].

Our results demonstrate the importance of social support as a potential protective factor. Social support has been the focus of many studies. In a study by Gottval et al. [27], social support was measured by the ESSI as in our study. It was shown that high social support is associated with low prevalence of PTSD. A plausible explanation is that of having a close person who listens and has a positive influence on the processing of traumatizing events at the memory level [42]. Intrinsic religiosity was equally important in our study as a potential protective factor of mental health, especially with regard to depression and generalized anxiety disorder. Similar results were found in other studies. It has also been reported [22,23,24] that religiosity has a positive impact on mental and physical health.

Turning to optimism, we see that, unlike other studies, our study shows a slight but significant correlation between optimism and PTSD. It could be due to the fact that the participants of our study are younger compared to the studies mentioned above and younger people are in general more optimistic than older generations [43].

Overall, our assumptions regarding protective factors could be confirmed, especially regarding the positive association of social support and religiosity with mental health. Significant results were also found concerning the feeling of being welcome in Germany and its positive association with mental health. This shows the importance of anti-discrimination measures. Future research should investigate the degree of perceived discrimination in everyday student life.

### Strengths and Limitations

Our study is the first in Germany that focused on the mental health of Syrian university students. Our investigation has several strengths: A high proportion of the registered students participated in the study. As it is usually very difficult to motivate refugees to participate in scientific surveys, this is a crucial advantage of our study. Additionally, the gender distribution of our sample is representative of the Syrians enrolled at FAU. The items of the questionnaire could be answered either in German or in Arabic to ensure that the answers were not distorted due to language comprehension problems.

Despite these strengths, there are also some limitations in our study: As it is well known, questionnaires are not the gold standard for diagnosing mental health problems. In addition, the results of our sample cannot be generalized to all Syrian students in Germany, and, thus, are not representative. Therefore, a study on a national level is recommended to replicate the results. Another limitation is the cross-sectional design of the study; thus, no causal conclusions can be drawn for mental health. Future studies should verify the correlations in a prospective design. Another limitation and, at the same time, an approach for further research projects is that in our study there was no control group of either German students or other countries for comparison. To ensure whether Syrian students are more stressed than Germans or other migrant students, such a comparative study would be necessary. Such an analysis could show whether studying, as a distress factor, may play a greater role in mental health or the refugee/migration background with its associated stressors.

## 5. Conclusions

The central finding of this study is that Syrian students have even higher burdens than the general Syrian population in Germany, except for PTSD. This suggests that besides the stress caused by flight and trauma, stressors such as studying and social isolation might be additional impediments for mental health and require intervention measures. This assumption should be investigated in studies of larger scope. Equally important is the finding that Syrian students who tested positive for mental distress reported significantly less social support and less optimism than participants without mental distress. The results of this study highlight the importance of protective factors, such as feeling welcome in the host country, social support, intrinsic religiosity and optimism, to reduce and avoid mental distress among Syrian students in Germany. To consolidate these findings, future studies should include control groups from the host society, as well as from other migrant populations. Further research would also be important to determine whether findings about Syrian students in Germany are also valid in the context of the European area or outside of it. Our hypothesis would be that one would come to similar conclusions in the European area, especially in Northern and Western Europe, because the structure and functioning of society are relatively similar. Outside the European area, a closer look at local conditions is needed.

As a practical implication, Syrian students should be provided with psychoeducation about psychological exhaustion, the importance of protective factors and about the healthcare structure in Germany, e.g., about special consultation hours at the university. Easily accessible psychosocial interventions should be developed and disseminated/introduced to students.

## Figures and Tables

**Table 1 ijerph-19-16200-t001:** Sample characteristics (socio-demographic and migration-specific variables).

Sample Characteristics	N	%
Age		
19–25	64	47.05
26–30	58	42.64
31–37	14	10.20
Sex		
Female	36	26.50
Male	100	73.50
Marital status		
Single	103	75.70
In a relationship	14	10.30
Married	19	14.00
Residence title		
Permanent residence permit	21	15.44
Temporary residence permit	46	33.82
Asylum entitlement/refugee status	45	33.08
Tolerated residence	0	0
Student visa	18	13.24
Other (subsidiary protection)	2	0.47
Secondary activities		
German classes	6	4.51
Integration classes	1	0.82
Voluntary activities	16	11.76
Side job	76	57.11
Internship/traineeship	7	5.33
No secondary activities	45	33.81
German language skills		
Beginner A1	1	0.70
Elementary A2	2	1.54
Advanced language use B1	1	0.73
Independent language use B2	9	6.73
Professional skills in the language C1	97	72.44
Near-native knowledge of the language C2	24	17.92
Religion		
Islam	109	80.14
Christianity	10	7.35
None	14	10.29
Other	2	1.47
Validity of residence permit		
Less than half a year	17	15.32
Less than a year	29	26.15
1–2 more years	60	54.13
3–4 more years	4	3.61
More than 5 years	1	0.72

**Table 2 ijerph-19-16200-t002:** Optimism, social support and religiosity: Comparison between mentally disordered and non-mentally disordered study participants.

	Total N = 135			Positive for Mental Distress ^1^ *n* = 58		Negative for Mental Distress ^2^ *n* = 77		Comparison *t*-Test
	M	SD	Range	M	SD	M	SD	2-sided *p*-value
Religiosity	8.23	3.26	3–12	8.07	3.18	8.36	3.34	0.62
Social support	18.54	5.14	5–25	16.95	5.43	19.79	4.52	0.001
Optimism	5.26	1.39	1–7	4.84	1.59	5.60	1.11	0.002

^1^ Subjects whose score exceeded the defined cut off point for depression, generalized anxiety disorder and/or PTSD were considered positive for mental distress. ^2^ Subjects whose score did not exceed the defined cut off point for depression, generalized anxiety disorder and/or PTSD were considered negative for mental distress.

**Table 3 ijerph-19-16200-t003:** Multiple linear regression analyses for depression, generalized anxiety and PTSD symptoms as dependent variables (N = 135).

Significant Predictors of Depressive Symptoms ^1,^*	B ^4^	SE ^5^	β	*p*	*95% CI* ^6^
Age	0.15	0.04	0.26	<0.001	0.07 to 0.23
Gender ^7^	−0.79	0.33	−0.18	0.02	−1.44 to −0.14
The feeling of being welcome in Germany	−0.22	0.06	−0.28	<0.001	−0.34 to −0.10
Intrinsic religiosity ^8^	−0.07	0.02	−0.23	0.002	−0.03 to −0.12
Social support ^9^	−0.07	0.03	−0.18	0.03	−0.13 to −0.01
Optimism	−0.17	0.11	−0.13	0.13	−0.39 to 0.05
**Significant predictors of GAD symptoms ^2,^***					
Age	0.03	0.05	0.06	0.46	−0.06 to 0.12
Gender ^7^	−0.62	0.36	−0.14	0.09	−1.34 to 0.10
The feeling of being welcome in Germany	−0.20	0.07	−0.24	0.004	−0.33 to −0.06
Intrinsic religiosity ^8^	−0.06	0.02	−0.18	0.02	−0.01 to −0.11
Social support ^9^	−0.09	0.03	−0.24	0.01	−0.16 to −0.03
Optimism	−0.17	0.12	−0.12	0.16	−0.42 to 0.07
**Significant predictors of PTSD symptoms ^3,^***					
Age	0.002	0.02	0.01	0.90	−0.03 to 0.03
Gender ^7^	−0.23	0.13	−0.16	0.08	−0.50 to 0.03
The feeling of being welcome in Germany	−0.0002	0.03	−0.001	0.99	−0.05 to 0.05
Intrinsic religiosity ^8^	0.01	0.01	0.09	0.29	−0.01 to 0.03
Social support ^9^	−0.04	0.01	−0.35	<0.001	−0.07 to −0.02
Optimism	−0.09	0.04	−0.19	0.05	−0.18 to 0.0004

* The following variables were included in multiple regression analyses: age, gender, the feeling of being welcome in Germany, the length of stay in Germany, social support, religiosity and optimism. In this table, only significant factors are shown. ^1^ PHQ-9: Patient Health Questionnaire—Depression Module, sum score, range: 0–6 (higher values are linked to higher psychological distress); ^2^ GAD-7, Generalized Anxiety Disorder Scale, sum score, range: 0–6 (higher values are linked to higher psychological distress); ^3^ IES: Impact of Event Scale, sum score, range: 0–24, (higher values are linked to higher psychological distress); ^4^ B: regression coefficient; ^5^ SE: standard error; ^6^ CI: confidence interval; ^7^ 1: female; 2: male; ^8^ DUREL: The Duke University Religion Index subscale sum score for intrinsic religiosity; ^9^ ESSI: ENRICHD Social Support Instrument; and PTSD: post-traumatic stress disorder.

## Data Availability

The data presented in this study are available on request from the corresponding author. The data are not publicly available due to privacy reasons.

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
