# Peer review of "Which Protective Factors Are Associated with the Mental Health of Syrian Students in Germany? A Register-Based Cross-Sectional Study"

_ijerph, 2022, doi:10.3390/ijerph192316200_

Round 1
Reviewer 1 Report
The following comments need to be addressed.
1. The abstract does not mention what methodologies were used.
2. The abstract needs to mention the results in an elaborative way.
3. The methods section needs a detailed description of the methods used.
4. The “Feeling to be welcome in Germany” has a P value of -0.01. Kindly advise.
5. Were other demographic variables, such as race and medical history considered in the model?
6. Provide the details about the outcome variable and the categorical variables used in the model.
7. Were the categorical variables in the model treated as categories or continuous variables?
Reviewer 2 Report
The article entitled “Which protective factors are associated with the mental health of Syrian students in Germany? A register-based cross-sectional study” contains an interesting and valuable correlational study regarding the relationship between protective factors and mental health in a largely unstudied student population. In my modest opinion, the study is of interest to a broad readership. With a few modest changes, it merits publication. The following are a few changes that may add clarity to the content of the paper:
In the introductory section, the definition of “protective factor” should be mentioned. What are its defining features?
In the last paragraph of the introductory section, the authors state that “the aim of our study was to investigate the frequency of mental disorders among Syrian students and the related protective factors. Another aim was to compare subjects with vs. without clinically relevant mental stress symptoms regarding their protective factors”. I think it will help readers to understand the study if the authors state the hypotheses underlying each aim.
At the start of the method section, the authors may include a summary subsection in which each aim and related hypothesis is followed by a brief description of the analysis that will be used to test the hypothesis.
In Table 3, all predictors should be displayed even though they do not make a significant contribution.
Regression analysis demands consideration of the sample size to obtain a reliable regression model. There are a lot of rules of thumb about this issue. One of them is that one should have at least 10-15 cases for each predictor in the model. See
Andy, F. (2009). Discovering statistics using SPSS. Sage
Section 3.4 is labeled “[o]ptimism, social support, and religiosity: Comparison between subjects with vs. without clinically relevant symptoms”. Comparisons for each of these variables are performed through a t-test. Would the data treatment make more sense if it is performed by running a logistic regression analysis in which optimism, social support, and religiosity are the predictors, and health status is treated as a binary outcome variable?
The breadth of the discussion section is rather limited. Thus, a subsection should be devoted to the implications and applications of the current results. Some consideration should also be given to the extent to which the current findings generalize to other populations of students who are studying abroad. Additionally, are Germany and other countries inside and outside the European community likely to present the same outcomes?
The paragraph that discusses the limitations of the study is considerably short. Are those mentioned by the authors the only limitations or weaknesses of the study?
The breadth of the section devoted to conclusions is rather incomplete. Can the authors consider combining a concise summary of their findings, the main applications and implications of the findings, and ideas for future research?
Round 2
Reviewer 1 Report
No further comments on my end.